# Essential Oils in Livestock: From Health to Food Quality

**DOI:** 10.3390/antiox10020330

**Published:** 2021-02-23

**Authors:** Ralph Nehme, Sonia Andrés, Renato B. Pereira, Meriem Ben Jemaa, Said Bouhallab, Fabrizio Ceciliani, Secundino López, Fatma Zohra Rahali, Riadh Ksouri, David M. Pereira, Latifa Abdennebi-Najar

**Affiliations:** 1Quality and Health Department, IDELE Institute, 149 rue de Bercy, 75595 Paris CEDEX 12, France; ralph.nehme@idele.fr; 2INRAE, Institut Agro, STLO, F-35042 Rennes, France; said.bouhallab@inrae.fr; 3Instituto de Ganadería de Montaña (CSIC-Universidad de León, Finca Marzanas s/n, 24346 Grulleros, Spain; sonia.andres@eae.csic.es (S.A.); s.lopez@unileon.es (S.L.); 4REQUIMTE/LAQV Laboratory of Pharmacognosy, Department of Chemistry Faculty of Pharmacy, University of Porto R Jorge Viterbo Ferreir 228, 4050-313 Porto, Portugal; rjpereira@ff.up.pt (R.B.P.); dpereira@ff.up.pt (D.M.P.); 5Laboratory of Aromatic and Medicinal Plants, Biotechnology Center of Borj-Cédria, Hammam-Lif BP 901 2050, Tunisia; mariembenjemaa@yahoo.fr (M.B.J.); fatma1707@gmail.com (F.Z.R.); ksouririadh@gmail.com (R.K.); 6Department of Veterinary Medicine Università degli Studi di Milano, 20122 Milano, Italy; Fabrizio.ceciliani@unimi.it; 7Departamento de Producción Animal, Universidad de León, 24007 León, Spain; 8Centre de Recherche Saint-Antoine (CRSA), Sorbonne University, INSERM UMR_S_938, 75020 Paris, France

**Keywords:** essential oils, antioxidant, antimicrobial, livestock, encapsulation, inflammation, mastitis, immunity, fermentation, nutrition

## Abstract

Using plant essential oils (EOs) contributes to the growing number of natural plants’ applications in livestock. Scientific data supporting the efficacy of EOs as anti-inflammatory, antibacterial and antioxidant molecules accumulates over time; however, the cumulative evidence is not always sufficient. EOs antioxidant properties have been investigated mainly from human perspectives. Still, so far, our review is the first to combine the beneficial supporting properties of EOs in a One Health approach and as an animal product quality enhancer, opening new possibilities for their utilization in the livestock and nutrition sectors. We aim to compile the currently available data on the main anti-inflammatory effects of EOs, whether encapsulated or not, with a focus on mammary gland inflammation. We will also review the EOs’ antioxidant activities when given in the diet or as a food preservative to counteract oxidative stress. We emphasize EOs’ in vitro and in vivo ruminal microbiota and mechanisms of action to promote animal health and performance. Given the concept of DOHaD (Developmental Origin of Health and Diseases), supplementing animals with EOs in early life opens new perspectives in the nutrition sector. However, effective evaluation of the significant safety components is required before extending their use to livestock and veterinary medicine.

## 1. Introduction

The intensification of animal husbandry and the globalization of trade have contributed to the extensive and improper use of antibiotics to combat infectious diseases. Since the end of the 1990s, awareness of this risk increased, prompting a gradual mobilization of the international community [1]. It has become an important issue as it has been recognized as a selective pressure, driving the accelerated emergence of bacterial resistance worldwide [2,3]. Prudent use of antibiotics in the future is urgently needed. It should be addressed in a One Health perspective, given that antibiotic resistance in humans, food, environment, and animals are connected, and exchange may continuously occur [4,5]. 

In 2006, the European Union banned all antibiotics as animal growth promoters and proposed alternatives [6]. Natural products, such as medicinal plants, essential oils (EOs), and herbal extracts, are regarded as promising alternative agents [7]. EOs are among the most economically relevant plant-derived products, being frequently responsible for several species’ health-promoting properties. These compounds are extracted from several plants generally localized in temperate to warm countries, like the Mediterranean and tropical countries, representing an important part of the traditional pharmacopoeia. EOs are a mixture of low-molecular-weight molecules that include terpenes (monoterpenes and sesquiterpenes), alcohols, aldehydes, and ketones which, in addition to their bioactive molecules, are also responsible for the aromatic fragrance that these materials frequently exhibit [8]. Presently, over 3000 EOs are known, 10% of which are commercially and economically relevant. These products are potential reservoirs of many bioactive compounds with several beneficial properties, and are aligned with current consumer preference for natural products [9]. They also offer the advantage of being better tolerated in the human body with fewer side effects [10].

EOs were also exploited for their flavoring properties [11,12] and for their potential to be used as food preservatives as they extend the length of the shelf life of products [13] or reduce the concentration of *Clostridium* numbers [14]. However, the main obstacle to using them as food preservatives is that they are often not potent enough and cause organoleptic alteration when added in sufficient quantities to induce the antimicrobial effect. However, this approach requires more insights into the interaction between molecules to determine the most synergetic effects, which remain elusive.

The popularity of EOs in the animal field and health has increased rapidly during the last decade. Nevertheless, their antioxidant and beneficial immunomodulatory effects in the ruminant sector remain poorly explored [15,16]. Mostly in poultry and pig sectors, studies have highlighted the capacity of some EOs to increase animal performances and efficiency by enhancing digestive secretions [17], increasing the number of probiotic bacteria such as *Lactobacillus* spp. [18,19], stimulating the immune function and the gastrointestinal microbiota [20,21], and decreasing oxidative stress [20]. However, some inconsistencies regarding the efficacy of EOs still exist, mainly linked to the nature of the compounds and some intrinsic and extrinsic factors, such as infection, nutritional status, environment, and particularly, diet composition [22]. 

The potential use of EOs as new nutraceutical additives during early growth (pregnancy and/or lactation) has not been explored yet and might be a very promising way to limit the use of antimicrobials in livestock, according to the DOHaD (Developmental Origin of Health and Diseases) concept. This concept was initially developed in humans by Barker [23,24] and was extended more recently to animal science [25,26]. This plan may raise early animals in an antibiotic-free production environment and drive heard performances and efficiency at later life, contributing to reduce antimicrobial resistance.

To our knowledge, little is known about the potential use of EOs in livestock. The literature’s significant findings are related to medicinal plant species’ effects among livestock disease without distinguishing the effects of their isolated bioactive compounds. Data on EOs in ruminant and particularly the dairy sector are also limited. Our review is the first to overview the current knowledge about the antioxidant and antimicrobial properties of EOs in both monogastric and ruminant husbandry, to treat mastitis, and use as feed additives, modulation of rumen metabolism, and in animal product-derived foods. Its finality is to drive the nutritional and pharmaceutical industries to the potential use of EOs to improve animal health and explore encapsulated EOs as enhancers of milk and meat quality products that can facilitate their implementation as natural preservatives in foods.

## 2. Methods

To cover a maximum range of publications from Scopus, Web of Science (WOS), and Google scholar, we screened for the different database terms “essential oils”, “antibacterial”, “anti-inflammatory”, “immunity, “additive”, “diet”, “animal products”, “livestock”, “encapsulation”, and in some cases, combining the primary taxonomic designation and botanical synonym names of plants with the listed key terms. Then, we performed an additional literature search from WOS on EOs in livestock. The total number of articles treating the use of EOs linked with animal production and livestock and their abstract between 2010 and 2021 was 919. The distribution of the number of papers according to the year of publication (2010–2021) is represented in Figure 1.

Among these articles, 138 records were obtained related to the characteristics of EOs as an anti-inflammatory, antibacterial, and immunoregulator, and 28 papers linking the use of EOs to treat mastitis, which is a new field of research.

Figure 2 shows an “Epic epoch” graph allowing to visualize the first 5 most used keywords in each article for each year and their evolutions. The map presents the different fields where EOs were used. We notice the growing evolution of the terms “chicken” and “storage” and the decreasing evolution of the word “diet”. 

Additional literature screening was performed to compare the number of articles with EOs in both monogastric and ruminants. When utilizing the keyword “EO” AND (“milk OR cow OR dairy OR cattle OR ruminant OR rumen OR sheep OR goat”), we obtained 478 articles between 2010 to 2021, with 15 reports in 2010, reaching a maximum of 80 papers in 2020. In non-ruminants, when screening with “EOs” AND (“pork OR pig OR swine OR chicken OR poultry OR monogastric”), we obtained 666 results with 40 articles in 2010 versus 110 in 2020. Note that the number of articles related to ”EO” AND “in vitro” studies in livestock animal was 118, while only 15 papers were related to in vivo studies.

## 3. Importance of EOs

EOs are known for their natural protecting role for the host plants and the fact that these oils contain properties many times more powerful than those found in dried herbs. Among these properties are the antibacterial, antimicrobial, antiviral, and antifungal activities, together with some particular medicinal effects that make EOs of very considerable interest [27,28].

### 3.1. Definition of EOs

EOs, also called volatile or ethereal oils [29], are aromatic lipophilic liquids obtained from plant material (flowers, buds, seeds, leaves, twigs, bark, herbs, wood, fruits, and roots). They can be obtained by expression, fermentation, enfleurage, or extraction, but steam distillation is most commonly used for commercial production of EOs [30]. The term ‘essential oil’ is thought to derive from the name coined in the 16th century by the Swiss reformer of medicine, Paracelsus von Hohenheim, who named the effective component of a drug Quinta essential [29]. As mentioned above, an estimated 3000 EOs are known, of which about 300 are commercially important destined chiefly for the flavors and fragrances market [30]. It has long been recognized that some EOs have antimicrobial properties [29], and these have been reviewed in the past [31,32], like antimicrobial properties of spices [31]. Still, the relatively recent enhancement of interest in ‘green’ consumerism has renewed scientific interest in these compounds [32,33]. Besides antibacterial properties [32,34,35], EOs or their components have been shown to exhibit antiviral, antimycotic [34], anti-toxigenic [35,36], antiparasitic [37,38], and insecticidal properties [39,40]. These characteristics are possibly related to these compounds’ function in plants [29,41].

### 3.2. Historical Use of EOs

Although spices were used for their perfume, flavor, and food preservatives since antiquity [42], only turpentine oil was mentioned in Greek and Roman history [29]. 

Concerning EO extraction, distillation was the first used method in the East (Egypt, India, and Persia) [29] more than 2000 years ago, and was improved in the 9th century by the Arabs [42].

The Catalan physician Villanova (ca. 1235–1311) wrote the first authentic EO distillation account [29]. It was only by the 13th century that pharmacies made EOs and pharmacopoeias described their pharmacological effects [42]. Still, EO’s utilization was not widespread in Europe before the 16th century via their trade in London City [43]. In that century, Brunschwig and Reiff, Strassburg physicians, mentioned, in each of their publications on EO distillation and use, only a few oils: anise, cinnamon, clove, juniper wood, mace, nutmeg, rosemary, spike, and turpentine [29]. The tea tree oil used for medical purposes has been recorded since Australia’s colonization at the end of the 18th century [44]. The first experimental measurement of EO antimicrobial efficiencies was performed in 1881 by De la Croix [24].

### 3.3. Current Use of EOs

Presently, EOs are used in various fields. The pharmaceutical, cosmetic, and food industries are increasingly interested in EOs due to their biological activities, notably antibacterial, antifungal, and antioxidant ones. For example, in the European Union (EU), EOs are the most used in food as flavoring agents, in cosmetics such as aftershaves and perfumes, and in pharmaceuticals for their attractive functional efficiencies [45]. Moreover, the pure molecules of EOs have been used as food flavorings [46], as dental care products [47], as antiseptics [48,49], and as feed supplements for livestock [50,51]. 

Natural preservatives based on EOs have become commercially available, such as ‘DMC Base Natural’. This food preservative (produced by DOMCA S.A., Alhendı’n, Granada, Spain) is composed of 50% EO mixture (rosemary, sage, and citrus) and 50% glycerol [52]. Another example of commercialized EOs is ‘Protecta I’ and ‘Protecta II’, consisting of blended herb extracts proposed by Bavaria Corp. Apopka, FL, USA. The contained EOs are dispersed in sodium citrate and sodium chloride solutions [53].

## 4. Extraction Methods for EOs

Extraction is the main factor that ensures the quality of Eos, and inappropriate extraction methods can cause losses in volatile compounds and decrease the quality of obtained extracts. There are two main categories of EOs’ extraction procedures: conventional and unconventional or innovative methods.

### 4.1. Conventional Extraction Methods

In this category of extraction techniques, the heating process’ water distillation is classically used to obtain EOs from aromatic plants by two methods: hydro distillation and steam distillation. As for the first method, plant material is immersed into boiled water inside the alembic and vapors are condensed onto liquid in a decanter separating the EO from water. In contrast, in the steam distillation, plant material is heated by water steam. Another method used to recover EO compounds is solvent extraction by using acetone, petroleum ether, hexane, methanol, or ethanol, but the inconvenience of this method is that it allows extracting other non-volatile compounds at the same time which makes the second step of EO purification essential [54].

### 4.2. Innovative Extraction Methods

New EO extraction techniques have emerged with many advantages in recent years, such as reducing extraction time, energy consumption, and solvents used and carbon dioxide emission [55]. Among these innovative techniques is solvent-free microwave extraction (SFME), a combination of heating plant samples using microwave energy followed by dry distillation at atmospheric pressure in the absence of any solvent. This method’s advantages are to obtain EOs with high yield and selectivity, shorter extraction time, and a green and safe process [56]. Another emerging method is supercritical fluid extraction using fluids at critical temperature (Tc) and pressure (Pc), having remarkable properties such as low viscosity, high diffusivity, and density closer to liquids [55]. Generally, for extracting EO compounds, carbon dioxide fluid is used, and the obtained extract is not pure and further purification stages are needed. Also, water use at subcritical state is a powerful method for extracting EOs [57]. The subcritical water stage is reached when water pressure becomes higher than the critical pressure and temperature lower than the critical temperature, or vice-versa. This extraction method is considered the best alternative EO recovery technique since it enables a fast isolation process at a low working temperature [58].

## 5. Chemical Composition of EOs

The composition of EOs from a particular species of plant can differ depending on harvesting seasons, genetics, agricultural practices, and geographical sources [59,60,61]. Such variation led to the formation of chemically different molecules, but with similarities. For example, in Greek oregano plants, the sum of cymene, γ-terpinene, carvacrol, and thymol is similar in EO derived from different geographic origins [62]. Also, it remains stable whatever the season of plant harvesting [63,64]. This indicates that the four listed compounds are biologically and functionally associated.

Adding fertilizers containing potassium nitrogen and phosphorous could increase the amount of EOs and change its composition. Fertilizers could increase the concentration of some enzymes responsible for the biosynthesis of terpenoids [61]. Sarmoun et al. showed that irrigation could affect the quality of EOs produced by *Rosmarinus officinalis* L.: the non-irrigated plant has a higher yield of EOs than the irrigated plant. However, the EOs, made from the irrigated *Rosmarinus officinalis,* show more variability in their compounds than the non-irrigated plant, such as trans-verbanol and linalyl-isobutyrate [65]. 

Generally, EOs produced from herbs harvested during or immediately after flowering possess the most potent antimicrobial activity [66]. Enantiomers of EO components have been shown to exhibit antimicrobial activity to different extents [67]. Table 1 shows the significant components of selected EOs that exhibit antibacterial properties.

## 6. Stability and Encapsulation of EOs 

Although EOs are considered secondary and non-essential plant metabolites, they have gained interest around the world due to their specific biological functions [61], many of which lend themselves to commercial exploitation. EOs production on a larger scale was started in the USA in the earlier 19th century [85]. Indeed, combining their established historical use as medicines to their range of biological activities aroused a great interest in their use as medicinal products [63]. Besides, attractive EOs, which leave a pleasant memory association, are used as marketing devices to sell cosmetic products, including detergents, scent perfumes, lotions, soaps, and household cleaners [86]. For instance, lavender EO and d-limonene from the citrus peel are potent solvents, and they are used in a wide variety of cleaning and cosmetic products [85]. For food industries, EOs are more and more considered to act as natural additives as antioxidant/antimicrobial agents [87]. The remarkable efficiency of most EOs against a wide range of pathogenic microorganisms, responsible for food spoilage and generally involved in food poisoning, was repeatedly reported [86,87,88].

However, it has been repeatedly discussed that EOs’ valorization in innovative industries is not economically and practically ideal [89,90]. One of the crucial issues with using EOs as green and efficient bioactive molecules is their impact on the final product’s sensory properties. The EO required to inhibit microbial growth may compromise organoleptic properties (aroma and taste) and produce an undesirable effect [91]. Alternatively, the use of lower concentrations of EOs may be possible if multiple preservation strategies that result in additive or synergistic effects on antimicrobial activity are involved [90,92].

Besides, it was confirmed that the incorporation of EOs in foods can be prejudice given their hydrophobicity and volatility [89]. Thus, EOs lose small quantities of volatile compounds when stored at high temperatures. Some components are highly unstable at pH variations, like citral, easily decomposed in an acidic environment [93]. Moreover, the hydrophobicity of EOs leads to their heterogeneous distribution in an aqueous medium. Such characteristics would lead to decreased EOs efficiencies as antimicrobial and antioxidant agents and limit their use as green and natural preservatives. 

Another valorization-limiting characteristic is EOs’ easy oxidation. Once deprived of the protective compartment in the plant matrix, EOs’ constituents are easily prone to different chemical transformations. Indeed, as EOs are both thermo-sensitive and volatile, they are easily alterable [94]. Several factors could be responsible for the degradation of the chemical constituent of EOs. First, the plant’s growth stage and the vegetable material’s post-harvest storage conditions profoundly influence an EO’s alteration. Besides, other factors related to handling or storing the oil itself will also lead to the acceleration of EOs’ oxidation, such as temperature, light, and atmospheric oxygen availability. Indeed, the ultraviolet light and visible light are considered accelerators of the self-oxidation process by taking out hydrogen, leading to alkyl radicals’ formation. The oxidation of terpenoids, for example, produces the *p*-Cymene (Figure 3). They also give rise to a range of stable oxidized products such as alcohols, ketones, and aldehydes [95].

The encapsulation of EOs seems to be an attractive new approach to overcome all previously listed impediments [90]. Encapsulation consists of coating sensitive materials (solid, liquid, or gas) known as core material, with a protective layer called wall material [96]. EO encapsulation is a complex process with interrelated steps [97]. In this context, several research groups focused on EO encapsulation, and different techniques were proposed [98]. Accordingly, choosing the encapsulation method and the encapsulating material depend on specific parameters: targeted mean particle size, the physicochemical characteristics of wall and core ingredients, the intended applications of the encapsulated material, the release system of capsule content, production capacity, and cost [96,99]. 

Concretely, the EO encapsulation approach has succeeded to convince the research society of its efficacy. Firstly, the oxidation phenomenon has been avoided/retarded after EOs’ encapsulation. Bakry et al. highlighted that basil EO’s microencapsulation protected it against oxidation under accelerated storage conditions for 49 days [100]. In addition, EO encapsulation allows the preservation of their biological efficiencies during their storage, transport, and processing. The encapsulation of oregano EO into microspheres of native sorghum and rice starch significantly prolonged its antioxidant activity during storage [101].

Moreover, using encapsulation as a technological alternative to conceal EOs’ highly undesirable taste has been repeatedly justified. Ghalem and Zouaoui suggested that adding *Lavandula* and *Chamaemelum* spp. EOs to yoghurt decreased its acceptability by panelists [102]. In addition, EO encapsulation represents an efficient approach to increase their bioactivity and bioavailability by ensuring their even distribution in the medium [103]. To overcome EO hydrophobic limitation, the outer protective layer, used in the encapsulation, acts as a hydrophilic barrier to EOs. In this context, Ben Jemaa et al. have pointed out that bulk thyme EO, with low water solubility, was roughly distributed in the medium and its action extent was therefore limited. On the contrary, when encapsulated in the nanoemulsion-based delivery system, its diffusion was homogeneously leading to more efficient antimicrobial activity [104].

In this context, multiple methods were developed for EOs’ encapsulation, as follows.

***The complex coacervation process:*** This process involves a liquid–liquid phase separation formed between two or more oppositely charged biopolymers from the initial solution, leading to coacervate formation around the active ingredient [100]. Mixtures between proteins and polysaccharides are generally used to prepare complex coacervates for encapsulation purposes. Encapsulation by complex coacervation can involve various steps for the formation of microcapsules: (i) an aqueous solution is prepared to contain two or more anionic and cationic polymers, (ii) the aqueous phase is mixed with a hydrophobic solution to produce a stable emulsion, (iii) a sudden change in pH and temperature induces the coacervation and the phase separation, and (iv) the polymer matrices harden by using a desolvating agent, high temperature, or a crosslinking agent (Figure 4) [105].

Recent studies showed that complex coacervation could be used by mixing two oppositely charged proteins with an affinity for an encapsulated bioactive component [106]. Efficient encapsulation yield with a protective effect was reported for vitamin B9 encapsulated in milk-derived protein coacervates. Complex coacervation is a mix of food proteins that constitutes a new, biosourced, and economically viable method suitable for entrapment and encapsulation of hydrophobic molecules such as EOs [106]. 

***Liposomes:*** Liposomes are made with many phospholipids bilayers vesicles enclosing an internal aqueous volume [107]. The growing interest in liposomes can be attributed mainly to their property of mimicking biological cells. Accordingly, liposomes are highly biocompatible, making them an ideal candidate for a drug delivery system, with applications ranging from delivering enzymes, antibacterial drugs, fungicides, and adjuvants for vaccines [108]. The membrane structure can vary significantly, making it possible to create different liposomes with diverse characteristics and applications [109].

***Nanoemulsion:*** In nanoemulsion, tiny EO droplets are confined to a cavity surrounded by a unique polymer membrane [110]. The EO dispersion in water is obtained through the chemical action of surfactants. It is highly recommended to encapsulate EO into nanoemulsion-based delivery systems for different reasons: non-toxic and non-irritant, high kinetic stability, capacity to solubilize hydrophobic bioactive molecules and to enhance their bioavailability, and suitability for human and veterinary uses [111]. The nanoencapsulation of EOs can be performed using multiple techniques, such as high-pressure homogenization [104], micro-fluidization [112], and sonication [113]. 

***Solid lipid nanoparticles (SLN):*** These particles are made up of lipids (triacylglycerols or waxes), solid at room temperature, biodegradable, and non-toxic, stabilized by adding suitable surfactants [97]. Solid lipids instead of oils allow improving the control in releasing the active molecules due to its reduced mobility in a solid phase compared to the oily liquid phase [107]. They are generally obtained by preparing an oil in water emulsion at a temperature exceeding the lipid phase melting point, and then the cooling of produced emulsion, leading to crystallization of the lipid and the trap of the target molecules either in the core or on the wall of the obtained particles [97]. This technique is highly recommended in carrying lipophilic active components since it offers adequate protection, less leakage, and sustained release of EOs [114]. 

## 7. Main Properties of EOs 

The therapeutic properties of EOs are directly linked to the molecules that compose them. They present antioxidant, antimicrobial, and anti-inflammatory properties.

### 7.1. Antimicrobial Activity

Several studies tried to explain the mechanism of action of an EO towards bacteria. This mechanism’s complexity is related to the chemical composition of EOs, which presents a high diversity of molecules (each molecule acts on a specific target). EOs can target the cell membrane, disrupting the cell energy status (the process of energy transduction coupled with the membrane and the solute transport) and the metabolic regulation. Sometimes, it modifies the expression of operons by inhibiting self-inducing mediators [115].

Many mechanisms of action are attributed to the interaction of EOs with the components of the cell membrane. The lipophilic character of molecules in EOs make them capable of penetrating the phospholipid double layer of the cell membrane, leading to a lack of regulation of the cell membrane, thereby disrupting the transportation of nutrients. Membrane transport can also dysregulate via disturbance of the ionic gradient on the cytoplasmic membrane’s two sides. However, some bacteria strains have been able to develop mechanisms allowing it to counteract this effect through ion pump use [115].

EOs could affect the biosynthesis of lipids, including unsaturated fatty acids, thus modifying the cell membrane structure due to the hydrophobic characteristic. EOs in the bacterial cell, even in a concentration below the minimum inhibitory concentration (MIC), decrease the level of unsaturated fatty acids that are generally responsible for the membrane fluidity. For example, thymol, carvacrol, and eugenol can increase saturated fatty acids and decrease C18 unsaturated fatty acids. The action of EOs can also affect the enzymes responsible for the latter’s biosynthesis [115].

Hundreds of in vitro studies were carried out to evaluate the effects of EOs on some pathogens, such as Gram-positive bacteria: *Staphylococcus aureus* and *Bacillus subtilis*, Gram-negative bacteria: *Pseudomonas aeruginosa* and *Escherichia coli*, yeasts: *Candida albicans*, and molds: *Aspergillus niger*. These six species are the most abundant infectious microorganisms, mainly because they are the most frequently tested [86]. Table 2 shows the minimum inhibitory concentration (MIC) (the lowest concentration of a chemical preventing a bacteria’s visible growth) of EOs against some pathogens [27]. 

### 7.2. AntiInflammatory Properties

Inflammation is a physiological response caused by an infected or injured tissue in the body. Inflammatory processes can: (i) increase the permeability of mucosal endothelial cells and the interstitium’s influx of blood leukocytes, (ii) cause the expression of cell adhesion molecules, such as vascular cell adhesion molecules (VCAM) and intercellular adhesion molecules (ICAM), (iii) upregulate the activity of several enzymes (oxygenase, peroxidase, and nitric oxide synthase) and the metabolism of arachidonic acid, and (iv) cause the release of pro-inflammatory cytokines [122,123,124,125].

EOs can interact with the signaling cytokines, the regulatory transcription factors, and the expression of pro-inflammatory genes in addition to their antioxidant activities. The mechanism of action may be indirect in an immunostimulatory reaction or direct via several mechanisms such as:

▪ Hyperemia, which accelerates the recruitment of leukocytes, promoting the anti-inflammatory reactions (citrals, citronellal, and cuminal in external use),

▪ Blockage of the synthesis and secretion of mediators of inflammation (histamine, pro-inflammatory cytokines, prostaglandins, leukotrienes, nitric oxide, free radicals), thus acting at different levels of anti-inflammatory activity [126,127]. 

## 8. EO and Inflammation: The Bovine Mastitis as a Paradigm Case

The immunomodulatory activity of EOs in human medicine has been reported for a long time, presenting the much more complicated evidence than previously believed, and not only limited to antioxidant activity. The EO obtained from *Lavandula angustifolia* showed the capacity to increase phagocytosis and decrease pro-inflammatory cytokine production, counter-balancing the inflammatory signaling induced by experimental infection with *S. aureus* [128].

In both ruminants and monogastrics, identifying immunomodulatory molecules for bovine mastitis, or calf, pig, and chicken intestinal diseases, is a promising alternative strategy to using antibiotics that may result in resistance development in humans and animals. Although there is a growing interest in the use of EOs, few data are available on the degradation rate of EOs and their compounds in the gastrointestinal tract and the way they are prepared and fed to animals, e.g., whether they are micro-encapsulated or not [129].

In monogastric species like pigs, the integration of diet with EO can change the distribution of lymphocytes in the gut. It is unknown whether this effect is direct, e.g., related to EOs’ interaction with the intestinal immune system, or indirect, via EOs’ interaction with the resident microbiome, which changes its relationship with the resident immune system. By integrating diets with a mixture of carvacrol, cinnamaldehyde, and capsicum oleoresin, it was possible to decrease intraepithelial lymphocyte (IEL) population jejunum and ileum and increase lymphocyte in the lamina propria of early-weaned pigs [130]. Indeed, this information does not allow to understand the consequence of this immunomodulatory activity for the overall animal health. On one side, a decrease of the immune activation may protect the animal from the unwanted excess of immune stimulation, and prevent collateral damages related to inflammation. On the other hand, decreasing immune activity might also pave the way for a breakthrough of pathogens.

On the contrary, other EOs’ activities are related to potentiating immune responses, such as improving serum lymphocyte proliferation rate, phagocytosis rate, immunoglobulin (Ig) G, IgA, IgM, C3, and C4 levels in piglets [131]. These results were recently confirmed by similar studies, demonstrating an increase of IgG and IgM after plant EO supplementation [132].

A recent study also reported the anti-inflammatory activity of encapsulated EO [133], demonstrated by a decrease of the major pro-inflammatory cytokines (Tumor necrosis factor alpha (TNF-α) and interleukin 6 (IL-6)). However, even anti-inflammatory cytokines, like IL-10, are decreased, confirming that EO activities are more complex, and not one-sided.

The molecular basis of the immunomodulatory activity of EOs has not been investigated in-depth and may indeed change, depending on the specific EO used. Regarding oregano-derived EO, the immunomodulatory activity is related to downregulation of pro-inflammatory pathways, including the mitogen-activated protein kinase (MAPK), protein kinase B (Akt), and nuclear factor κB (NF-κB) signaling pathways, which reflects on the decreased expression of inflammatory cytokines in the jejunum of pigs models [134].

The effects of dietary EO have been studied in the other primary livestock monogastric species, the chicken, albeit less intensively. The EOs have an evident immunomodulatory activity on mRNA expression in intestinal cells on IL-17A, Interferon (INF) α, INF-γ, Transforming growth factor (TGF) β, and IL-10 [135]. Carvacrol can decrease the inflammatory activity induced by lipopolysaccharides (LPS) treatment by downregulating the mRNA expression of some pro-inflammatory cytokines, including TNF-α, IL-1β, IL-6, IL-8, and of pathogen-associated molecular pattern membrane receptors, such as TLR2 and 4, and the downstream pathways related to them, such as NF-κB [136], after LPS challenge, demonstrating the potential anti-inflammatory activities of this EO against intestinal infection. A study on *Minthostachys verticillata* EO with immunomodulatory activity on human T cells and anti-inflammatory activities showed its capability to increase macrophage phagocytosis and oxidative burst in a Balb/c mice model. Moreover, this EO can decrease neutrophil chemotaxis to the mammary gland, and downregulate pro-inflammatory cytokines’, such as IL-1β and TNF-α, mRNA expression. Consistently, the anti-inflammatory cytokine IL-10 is also upregulated. Given that *M. verticillata* EO was also shown to induce a decrease of the bacterial count in the glands of mice experimentally infected with *E. faecium*, this model demonstrated that this EO could boost the innate immunity to help to resolve inflammation [137]. 

In ruminants, the anti-inflammatory activity of EO was also reported in a model of Sub-Acute Ruminal Acidosis: introducing a mixture of EO and polyphenols in the diet could reduce the number of circulating neutrophils and of the acute phase proteins Serum Amyloid A, Haptoglobin, and Lipopolysaccharide Binding proteins, that are regarded as markers of inflammation in cows [138,139].

The mammary gland’s inflammation, also called mastitis, is one of the most frequent and costly dairy cows’ diseases, mostly caused by intramammary infection due to bacteria and other microorganisms [140]. Bovine mastitis is an inflammation of the udder. It is the result of an infection of the udder by bacteria, yeasts, or viruses. Mastitis or inflammation of the mammary gland develops when a pathogen crosses the teat canal barriers and multiplies in the milk. When the immune defense mechanisms fight this infection quickly and effectively, mastitis will be mild and transient. On the other hand, when the defense mechanisms are compromised, during parturition, or when the pathogen has evasion mechanisms against the immune system, mastitis will be more severe or become chronic [141].

The intensity of the inflammatory response will determine the type of mastitis: subclinical or clinical. Major pathogens related to mastitis include *Escherichia coli, Staphylococcus aureus, Streptococcus uberis, Streptococcus dysgalactiae*, and *Streptococcus agalactiae,* and other bacteria, such as *Arcanobacterium pyogenes, Staphylococcus* non-aureus, *Pseudomonas aeruginosa, Nocardia asteroides, Clostridium perfringens, Mycobacterium* spp., *Listeria monocytogenes,* and many others can be found as associated with mastitis. Following clinical signs, bacterial infiltration reaction in dairy cows develops following two different clinical events: clinical mastitis and subclinical mastitis [142]. The clinical mastitis usually recapitulates acute inflammation’s significant features: the infected udder becomes swollen, red, hot, and sometimes painful to be touched. Clinical mastitis significantly affects milk quality and quantity.

The resulting milk looks like an inflammatory exudate, hemorrhagic or sometimes purulent, exhibits the formation of flakes and clots, decreased protein, fat, and lactose contents. The exudate is infiltrated by cells migrated from the blood via diapedesis, mostly neutrophil granulocytes, and are defined as somatic cells (SC) [141]. Subclinical mastitis is more challenging to detect. In most of the cases, the only sign is the increase of SC. Bovine mastitis causes a drop in milk production in the affected quarters. This decrease is most marked in the event of clinical mastitis. Subclinical inflammations can also reduce milk productivity by up to 40%. As a result, veterinary costs, pharmacological treatment, and milk wastage due to treatments impact the farm’s profitability [143]. Beside the decrease in milk production in animals with mastitis, the milk containing somatic cells is removed from the dairy chain, increasing the economic burden [144]. 

The conventional therapy against mastitis still includes aggressive treatment with antibiotics which, although necessary, is not fully efficient, and presents several drawbacks. Extending over the critical use of antibiotics is at the background of the development of antimicrobial resistance that can persist in the bacterial community [145], as shown for *Streptococcus agalactiae* and *S. aureus* [146], causing antibiotic resistance in humans as well [142]. Alternative and complementary approaches are therefore investigated, such as using EOs. 

In vitro studies have demonstrated the antibacterial activity of EOs against several pathogens that can cause mastitis, such as, among the others, *L. monocytogenes*, *Salmonella typhimurium, E.coli,* and *S. aureus* [147]. The EOs’ chemical constituents promote bacterial cell wall and membrane damages, such as cell wall degradation, protein denaturation, and destabilization of proton motive force. Given their lipophilic nature, EOs can penetrate the lipid bi-layer of the bacterial cell membrane, causing loss of integrity and structural organization [148]. In general, Gram-positive bacteria are more sensitive than Gram-negative to disruptive activity, probably due to the lipopolysaccharides at the outside of the cell wall [149].

Their natural origin, coupled with lower side effects and the limited development of resistance after prolonged use, have identified EOs as promising therapeutics agents against mastitis, both in vitro and in vivo [150]. 

In vitro studies have been carried out to test the effects of selected EOs on the cells present in the mammary glands, such as epithelial mammary gland cells, or against bacteria involved in mastitis development. The activity of EOs against Methicillin-resistant *S. aureus* (MRSA) may be related to the disruption of biofilms in a syntrophic consortium within a self-produced matrix of extracellular polymeric substances (EPS). Biofilm production is a strictly coordinated process by which planktonic bacteria switch from free-floating forms to sessile anchored cells embedded in self-produced EPS [151]. 

Remarkably, several studies demonstrated that the major components of EOs, when isolated, appear to be less active against bovine pathogens than does the full-fledged EO, suggesting the hypothesis that, along with the individual antibacterial activity of each component, the various molecules present in the EO act synergistically to fulfil an antibacterial activity [148].

As expected, there are very limited and less conclusive in vivo studies. Moreover, in many cases, the molecules used for in vivo studies were not tested for the presence of lipopolysaccharides or other pro-inflammatory molecules, which may bias the experimental design, particularly when directly infused in the mammary gland. Most in vivo studies, including the treatment with external application, do not provide enough scientific evidence to support the use of the EO directly in the mammary gland.

On the other hand, as mentioned previously, bibliographic research highlights a low number of in vivo experiments made on the use of EOs in treating mastitis: Abboud et al. studied the effect of 10% of a mixture of *Thymus vulgaris* and *Lavandula angustifolia,* by intramammary infusion and external application on the quarter. After four days of treatment, they discovered a substantial decrease in the bacterial colony count. The most potent antibacterial activity was achieved by massaging the udder with the mixture of EOs [152].

Another study was conducted to evaluate the effect of *Origanum vulgare* by intra-mammary infusion: 0.9 mL of EO was given twice a day for three days. *S. aureus* and *E. coli* were not detected in milk after the treatment [153]. The intramammary infusion of sage EO to ewes affected with subclinical mastitis resulted in a significant decrease in somatic cell count 24 and 48 h post-treatment. On the other hand, Lefevre et al. treated 55 cases of mastitis with a mixture of EOs containing *Thymus vulgaris, Rosmarinus verbenone,* and *Laurus nobilis* (1.5% each in 10 mL of sunflower oil), and 45 mastitis with a combination of *Thymus vulgaris* and *Rosmarinus verbenone* (6% of each in sunflower oil or water). The results were unsatisfactory: The recovery rate was only 40% [154].

Several hypotheses were advanced trying to explain these contradictory observations: (i) milk can interact with the EO, resulting in an alteration of their antibacterial properties, (ii) the concentration of the EO used in the in vivo treatment is not enough, and (iii) the excipient could modify the physicochemical characteristics of EOs [155].

### Antioxidant Activity and Mechanisms of Action

EOs obtained from different plants are a source of natural antioxidants [156]. Most EOs have the virtue of being non-toxic—the literature has demonstrated that, compared to inorganic supplements, plant-derived products such as EOs are less toxic, residue-free, and thought to be ideal growth promoters for both milk and beef production and quality [156,157]. In addition, transition metals such as Fe and Cu, often given as mineral supplementation, favor the formation of highly reactive free radicals in meat, accelerating its oxidation [158,159]. However, in high concentrations, they can exert toxicity, such as necrosis. Hence, it is a common practice in the study of EOs, although not entirely correct, to identify natural antioxidants as “molecules able to react with radicals” or molecules that present the reducing power to counteract the oxidative stress caused by radicals [156,160]. Accordingly, there are several methods used to examine the antioxidant properties of EOs. The most common tests used to screen the antioxidant activity of EOs are based on the in vitro reaction of the phytochemicals with some colored persistent radicals (e.g., 2,2-diphenyl-1-picrylhydrazyl (DPPH^•^) or 2,2′-azino-bis(3-ethylbenzothiazoline-6-sulfonic acid) ABTS^•+^ assay) (antiradical) or with some oxidizing nonradical species such as Fe^3+^ ions (e.g., ferric reducing antioxidant power (FRAP) assay) [161]. From a mechanistic point of view, these methods are classified as electron transfer (ET)-based assays, contrary to hydrogen atom transfer (HAT)-based assays, also widely used, which include oxygen radical absorbance capacity (ORAC) assay, radical-trapping antioxidant parameter (TRAP) assay, crocin bleaching assay using 2,2′-azobis-2-methyl-propanimidamide dihydrochloride (AAPH) as a radical generator, and β-carotene bleaching (BCB) assay [162,163]. In Table 3, we summarized some studies on the *in vitro* antioxidant activity of the selected EOs. 

Many studies exploring the bioactivity of EOs have mainly attributed their antioxidant capacity to terpenoids with phenolic groups such as carvacrol, methyl chavicol, thymol, and eugenol, as they can donate hydrogen atoms to free radicals and transform them into more stable products [162,164]. For example, about 80% of oregano EOs are constituted by carvacrol and thymol, mainly responsible for its antioxidant activity [165,166]. Similarly, EOs from other aromatic plant species, like lemon balm, basil, thyme, and sage, have also been established as rich sources of antioxidants [162]. The study conducted by Viuda-Martos and co-workers demonstrated the ability EOs from oregano (*Origanum vulgare*), rosemary (*Rosmarinus officinalis*), and sage (*Salvia officinalis*) to chelate Fe^2+^, with rosemary EO displaying the highest effect (76.06%). Moreover, oregano EO showed the most increased antioxidant activity in the Rancimat test [167]. Tunisian *Thymus capitatus* EO’s antioxidant activity, mainly composed of carvacrol, *p*-cymene, and γ-terpinene, was compared with butylated hydroxyanisole (BHA) and Butylated hydroxytoluene (BHT) by DPPH and thiobarbituric acid-reactive species (TBARS) methods, evidencing better antioxidant properties [168]. On the other hand, Kulisic et al. revealed that the antioxidant activity of *O. vulgare* EO was less effective than ascorbic acid (AA), but comparable with the α-tocopherol and BHT, based on the BCB test, DPPH radical scavenging method, and thiobarbituric acid-reactive species (TBARS) assay [169]. Furthermore, in vivo experiments carried out in pigs fed with diets containing *O. vulgare* crude herbal drug promoted significantly improved stability and lower cholesterol oxide content in raw belly bacon than controls after 34 weeks of storage [170]. In line with these findings, Botsoglou et al. demonstrated that laying hens fed with a *Thymus vulgaris*-enriched diet (3% ground herb) led to an increased storage time (over 60 days) in the refrigerator, mostly due to reduced oxidation of liquid yolk [171]. Additionally, higher enzyme levels of the antioxidative enzymes, namely superoxide dismutase (SOD) and glutathione peroxidase (GPx), were found in ageing rats dietary supplemented with thyme oil or thymol [172]. Studies conducted in White pigs demonstrated that rosemary EO (150 mg/kg) significantly reduced the generation of TBARS and hexanal, and carbonyls, indicators of lipid and protein oxidation, respectively [165,173]. A study developed by Takayama et al. showed that pre-treatment of rats with *R. officinalis* EO (50 mg/kg) protected against the gastric injury induced by ethanol, probably by modulating the activities of SOD and GPx, and increasing or maintaining the levels of glutathione (GSH) [174]. In vitro experiments dealing with the DPPH scavenging activity of *Nigella sativa* EO revealed that its potent antioxidant activity (The half maximal inhibitory concentration (IC_50_) = 19 µg/mL) could be attributed to the presence of oxygenated monoterpenes such as thymol and thymoquinone [175]. Furthermore, *N. sativa* EO strongly inhibited tert-Butyl hydroperoxide (*t*-BuOOH) induced 2′,7′-dichlorofluorescin diacetate (DCFH) oxidation with an IC_50_ of 1.0 μg/mL in WS-1 cells, indicating its ability to inhibit reactive oxygen species (ROS) production [176]. In vivo assays demonstrated that *N. sativa* EO holds the potential to significantly improve the oxidant status of normal rats by enhancing the activity of several antioxidant enzymes, namely glutathione reductase and glutathione transferase [177]. Singh et al. also showed the strong antioxidant effect of *N. sativa* EO in the rapeseed oil system, evidencing comparable results to BHA and BHT effects at the 6 mg levels [158].

**Table 3 antioxidants-10-00330-t003:** Antioxidant activity of selected EOs.

Essential Oil (Plant Species)	Location (Plant Part)	Main Components	In Vitro Assay, Bioactivity	References
**Sage (*Salvia officinalis* L.)**	Saudi Arabia (leaves)	Camphor (20.3%), 1,8-cineole (15.0%), α-thujone (14.9%), viridiflorol (9.9%), carvone (6.2%), β-thujone (5.7%)	DPPH, IC_50_ = 970 µg/mL	[178]
	Morocco (leaves)	Camphor (24.1%), α-thujone (21.4%), 1,8-Cineole (16.5%), α-pinene (11.2%), camphene (6.9%)	DPPH, IC_50_ = 2.12 mg/mL (> Q and AA); BCB, RC_50_ = 3.78 mg/mL (> Q and BHT); FRAP, EC_50_ = 2.98 mg/mL (> Q and AA)	[179]
	Spain (leaves and flowers)	Camphor (25.0%), 1,8-cineole (24.7%), camphene (7.6%), α-pinene (6.8 %), α-terpinyl acetate (6.0%)	DPPH, IC_50_ = 4.20 mg/mL (> AA and BHT); TBARS, EC_50_ = 35.56 mg/mL (> AA and BHT); FIC, EC_50_ = 7.16 mg/mL (< AA and BHT); Rancimat, AAI = 1.07 (< AA and BHT)	[167,180]
	Tunisia (leaves)	Camphor (25.1%), α-thujone (18.8%), 1,8-cineole (14.1%), viridiflorol (8.0%), β-thujone (4.5%), and β-caryophyllene (3.3%)	DPPH, IC_50_ = 6.7 mg/mL (> BHT); FRAP, IC_50_ = 28.4 mg/mL (> BHT); LAP inhibition, IC_50_ = 9.6 mg/mL (> α-tocopherol)	[181]
**Laurel (*Laurus nobilis* L.)**	Morocco; (Flowers)	1.8-Cineole (45.0%), α-caryophyllene (7.5%), germacradienol (6.1%), limonene (4.7%), α-pinene (3.0%), germacrene D (3.1%)	DPPH, IC_50_ = 82.01 µg/mL (> BHT and AA)	[182]
	Turkey (leaves)	1,8-Cineole (51.8%), α-terpinyl acetate (11.2%), sabinene (10.1%), α-terpineol (5.2%)	DPPH, IC_50_ = 59.2 µg/mL (> BHT and AA); BCB, RAA = 76.8 % (< BHT and AA)	[183]
	Lebanon (leaves)	1,8-Cineole (35.2%), 1-*p*-menthen-8-ethyl acetate (13.5%), linalool (7.1%), sabinene (6.2%), α-pinene (5.7%)	DPPH, IC_50_ = 53.5 µg/mL (> AA); BCB, IC_50_ = 35.6–38.9 µg/mL (> PG)	[184]
	Lebanon (seeds)	β- Ocimene (21.8%), 1,8-cineole (9.4%), α-pinene (3.7%), β-pinene (2.1%)	DPPH, IC_50_ = 66.1 µg/mL (> AA); BCB, IC_50_ = 41.1–45.9 µg/mL (> PG)	[184]
**Coriander (*Coriandrum sativum* L.)**	Slovakia (fruit)	β-Linalool (66.1%), camphor (8.3%), geranyl acetate (6.9%), cymene (6.4%)	DPPH, IC_50_ ~ 39.38 mg TEAC/L	[185]
**Rosemary (*Rosmarinus officinalis* L.)**	Morocco (leaves)	1,8-Cineole (42.3%), camphor (14.8%), α-pinene (8.9%), β-pinene (6.3%), α-terpineol (5.6%), borneol (4.9%)	DPPH, IC_50_ = 4.82 mg/mL (> Q and AA); BCB, RC_50_ = 5.79 mg/mL (> Q and BHT); FRAP, EC_50_ = 5.62 mg/mL (> Q and AA)	[179]
	Serbia (leaves)	Limonene (21.7%), camphor (21.6%), α-pinene (13.5%), Z-linalool oxide (10.8%)	DPPH, IC_50_ = 3.82 µL/mL (< BHT)	[186]
	Serbia (aerial parts)	1,8-Cineole (43.8%), camphor (12.5%), α-pinene (11.5%), β-pinene (8.2%)	DPPH, IC_50_ = 77.6 μL/mL (= α-tocopherol)	[187]
**Conehead thyme (*Thymus capitatus*)**	Morocco (aerial parts)	*p*-Cymene (18.9%), carvacrol (13.4%), geranyl acetate (12.2%), borneol (10.2%)	DPPH, IC_50_ = 103 µg/mL (> AA)	[188]
	Tunisia (aerial parts)	Carvacrol (65.6–80.7%), *p*-cymene (4.8–8.9%), γ-terpinene (5.3–13.7%), β-caryophyllene (1.8–3.2%)	DPPH, IC_50_ ~ 250 µg/mL (< BHT); TBARS, IC_50_ < 100 µg/mL	[168]
	Algeria (leaves)	Thymol (51.2%), carvacrol (12.6%), γ-terpinene (10.3%)	DPPH, IC_50_ = 0.62 µg/mL (< BHA, AA, and thymol); FRAP, EC_50_ = 2.13 µg/mL (< BHA, AA, and thymol); TAC, EC_50_ = 0.78 µg/mL (< BHA, AA, and thymol)	[189]
**Black cumin** **(*Nigella sativa*)**	Austria (seeds)	Thymoquinone (30–48%), *p*-cymene (7–15%), carvacrol (6–12%), 4-terpineol (2–7%), longifolene (1–8%), *t*-anethole (1–4%)	DPPH, IC_50_ = 460 µg/mL (> BHT, AA, Q, carvacrol, and thymoquinone); ^•^OH, IC_50_ = 0.011 µg/mL (< carvacrol and thymoquinone)	[76]
	Iran (seeds)	Thymoquinone (20.3 %), camphene (11.0 %), thymol (10.1 %), β-pinene (7.0 %), α-thujene (6.0%), γ-terpinene (5.1 %)	DPPH, IC_50_ = 19 µg/mL (> BHA, AA, and thymoquinone, but < α-thujene, β-pinene, *p*-cymene, and γ-terpinene)	[175]
	Tunisia (seeds)	*p*-Cymene (60.5%), α-thujene (6.9%), γ-terpinene (3.5%), thymoquinone (3.0%), β-pinene (2.4%), carvacrol (2.4%), terpinen-4-ol (2.1%)	DCFH (ROS production), IC_50_ = 1 µg/mL (< Q)	[176]
**Cade (*Juniperus oxycedrus*)**	Tunisia (leaves)	β-Phellandrene (36.8%), α-terpinolene (13.2%), β-myrcene (9.1%), α-campholenal (9.0%), and *p*-cymene (6.0%)	DPPH, IC_50_ = 20.1 µg/mL (< BHT); FRAP, 15.9 μmol Fe^2+^/g	[190]
	Turkey (needles)	Caryophyllene oxide (31.6%), α-pinene (24.3%), caryophyllene (10.0%)	DPPH, IC_50_ < 40 µg/mL (< BHA and BHT); FRAP, 179.6 µmol TX/g (< BHA and BHT)	[191]
	Turkey (cones)	β-Pinene (29.9%), α-pinene (26.6%), limonene (9.7%)	DPPH, IC_50_ < 40 µg/mL (< BHA and BHT); FRAP, 415.38 µmol TX/g (< BHA and BHT)	[191]
	Algeria (aerial parts)	α-Pinene (37.8%), abietadiene (8.3%), bulnesol (7.2%), manoyl oxide (5.0%), germacrene D (4.8%)	DPPH, IC_50_ = 91.25 mg/mL (> AA); FRAP, 0.97 µmol TX/g (< AA); ABTS, 5.82 µmol TX/g (< AA)	[192]
**Geranium (*Pelargonium graveolens*)**	India (leaves)	Citronellol (37.0%), geraniol (18.0%), citronellyl formate (5.5%), linalool (4.1%), rose oxide (2.4%), geranyl formate (2.2%)	DPPH, IC_50_ = 18.02 μg/mL (> AA); NO, IC_50_ = 19.98 µg/mL (> AA)	[193]
	Egypt (leaves)	β-Citronellol (35.9%), geraniol (11.7%), citronellyl formate (11.4%), linalool (9.6%), (+)-isomenthone (6.4%), σ-selinene (5.5%)	DPPH, IC_50_ = 6.2 μg/mL (> BHT); BCLA, IC_50_ = 4.1 μg/mL (> BHT)	[194]
**Oregano (*Origanum vulgare*)**	Croatia (flowered tops and stalks)	Thymol (35.0%), carvacrol (32.0%), γ-terpinene (10.5.%), *p*-cymene (9.1%), α-terpinene (3.6%)	DPPH, IC_50_ = 0.5 mg/mL (> BHT, AA, and α-tocopherol, but = thymol); TBARS, AAI = 29.9% (< BHT, α-tocopherol, but = thymol and carvacrol)	[169]
	Chile (aerial parts)	Thymol (15.9%), *Z*-sabinene hydrate (13.4%), γ-terpinene (10.6%), *p*-cymene (8.6%), linalyl acetate (7.2%), sabinene (6.5%), carvacrol methyl ether (5.6%), carvacrol (3.1%)	DPPH, IC_50_ = 4.75 mg/mL (> Q and TX); ABTS, 1252.7 µmol TX/g; FRAP, 270.5 µmol TX/g	[195]
	Algeria (aerial parts)	*p*-Cymene (25.6%), thymol (23.1%), carvacrol (20.3%), γ-terpinene (16.6%), α-terpinene (1.8%)	DPPH, IC_50_ = 0.461 mg/mL (> AA)	[166]
**White wormwood (*Artemisia herba-alba*)**	Algeria (aerial parts)	Davanone D (49.5%), camphor (10%), *trans*-γ-cadinene (3.9%)	DPPH, IC_50_ = 2.61 mg/mL (> than AA); FRAP, 8.17 µmol TX/g (< AA); ABTS, 6.74 µmol TX/g (< AA)	[192]
	Morocco (aerial parts)	β-Thujone (42.9%), chrysanthenone, α-terpineol (9.7%), α-thujone (5.4%), α-pinene (4.6%), bornyl acetate (2.4%)	DPPH, IC_50_ = 2.9 µg/mL (> BHT)	[196]
	Tunisia (aerial parts)	α-Thujone (17.6%), *trans*-sabinyl acetate (17.2%), chrysanthenone (8.3%), β-thujone (7.6%), germacrene D (7.0%)	DPPH, IC_50_ = 1.17 mg/mL (> TX); FRAP, 161.6 µmol Fe^2+^/g; BCB, 0.6 mg/mL (> BHT)	[197]

DPPH: 2,2-diphenyl-1-picrylhydrazyl; ABTS: 2,2’-azino-bis(3-ethylbenzothiazoline-6-sulfonic acid); BCB: β-carotene bleaching; FRAP: ferric reducing antioxidant power; TBARS: thiobarbituric acid-reactive species; FIC: ferrous ion-chelating; BCLA: β-carotene-linoleic acid assay; AAI: antioxidant activity index; TAC: total antioxidant capacity; RAA: relative antioxidant activity; TEAC: trolox equivalent antioxidant capacity; LAP: linoleic acid peroxidation; ROS: reactive oxygen species; DCFH: 2′,7′-dichlorofluorescin diacetate; BHA: butylated hydroxyanisole; BHT: butylated hydroxytoluene; AA: ascorbic acid; Q: quercetin; PG: propyl gallate.

Care should be taken before assuming that the antioxidant property of EOs is simply that of one characteristic component. Identifying a leading antioxidant compound and the elucidation of the exact mechanism of action is a herculean task due to the presence of a plethora of compounds with different molecular structures of antioxidant powers. However, EOs’ chemical composition can allow to roughly predict its antioxidant potential, since it is expected that EOs containing higher amounts in phenolics and modest content in unsaturated terpenes display more robust antioxidant properties. One additional point should be considered concerning the antioxidant activity of EOs, and that is the enormous variability in chemical composition, even in EOs obtained from the same plant species, which hampers a direct comparison of published data. Such diversity is strictly connected to multiple factors, including geographic location, parts of the plant used, harvest time, environmental and agronomic conditions, phenological stage, and extraction methods [198].

Furthermore, synergistic or antagonistic effects could occur among the components, and such interactions may positively or negatively affect the ultimate antioxidant properties of EOs. For the aforementioned reasons, more in-depth knowledge of the mechanism of action and effects of individual compounds would allow the formulation of mixtures of compounds with optimized efficacy. Moreover, the vast amount of information being generated by in vitro assays requires further validation through systematic animal studies and clinical investigation.

## 9. EOs and Rumen Fermentation

EOs have been proposed for use in ruminant feed targeting different nutritional purposes and modulating the rumen’s anaerobic fermentation processes. Microbial digestion in the rumen can be improved by using feed additives. Some additives could drive the fermentation reactions towards the formation of more efficient products by acting on the balance of the microbial population. Antibiotics have been widely used in the past as microbiota regulators. However, their use as feed additives has been banned in the EU since 2006. In this context, EOs could offer some new opportunities to replace antibiotics [199].

Some of the effects of EO can be indirectly related to EO activity on gut microbiota. Most of the studies have reported EO’s effects on gut microbiota in pigs and poultry, whereas research on EOs’ impact on ruminant microbiota is somehow lagging. Feeding pigs with a blend of oregano and cinnamon oil increased *Lactobacillaceae* and *Ruminococcaceae* and decreased *Clostridiaceae* in the intestine [200]. *Firmicutes* and *Bacteroidetes* were the two most dominant phyla in a study involving a mixture of EOs and organic acids in weaned piglets, which also induced an increase in performance [201]. Dietary supplementation with EO was associated with increased Lactobacilli and decreased coliforms in vivo [202]. Feeding thymol- and cinnamaldehyde-supplemented diets induced a decrease in *E. coli* carriage in piglets at a similar level to that obtained with in-feed antibiotics [203]. These results are mostly consistent using, in most cases, culture-related techniques. Introducing next-generation sequencing approaches to study EOs’ impact on microbiota would probably provide more insights into the relationship between EO and the microbial community in the intestine and, in turn, how this relationship may shape the immune system. In poultry, treatment with EO has a synergistic effect with antibacterial peptides to reduce pathogenic intestinal microbiota’s negative impact. Indeed, cecal microbiota composition can be substantially changed with either cinnamaldehyde alone or in combination with citral in the feed [204]. It is commonly found that EO may exhibit a potential antimicrobial activity against bacteria, archaea, protozoa, or fungi in the rumen [205]. Patra described EOs’ potential to inhibit hyper-ammonia production bacteria, selective inhibition of *Selenomonas ruminantium* by thymol, and no substantial EO effects on ruminal ciliate protozoa. At high doses, EO can be highly inhibitory for most ruminal microorganisms [206].

However, other observations have shown that some compounds may not be inhibitory depending on the EO and the dose used. Using microarray analysis, Patra and Yu observed that some Firmicutes (especially *Clostridia*) and *Butyrivibrio* were decreased by EOs, whereas *Bacteroidetes* (*Prevotella*) were increased. However, the effects were highly dependent on the type of EO supplemented [207]. Oh and Hristov showed that dietary supplementation with oregano leaves in dairy cows decreased *Ruminococcus flavefaciens*, an important rumen fibrolytic bacteria [208]. However, Kim et al. observed an increase in several bacterial species (*Selenomonas ruminantium, Ruminococcus albus, Butyrivibrio fibrisolvens,* and *Ruminococcus flavefaciens*) and in fungi in response to an EO mixture [209]. Alagawany et al. reviewed the reported observations on effects of oregano and its main compounds (carvacrol, thymol) and found somewhat controversial effects.

In most cases, a depressing impact on methanogen and fungal counts was reported [210]. Ruminal protozoa can also be reduced in response to EO activity [211,212]. Metagenomic analyses on the ruminal fluid from goats receiving EO-supplemented diets demonstrated ruminal microbiota changes in a dose-dependent way [213].

Considering the multiple effects of EO on rumen microbiota, it has been proposed that EOs could improve the feed efficiency and nutrient utilization in ruminants. EO can also decrease protein degradation in the rumen, whereas controversial effects on fiber degradation have been reported. In some cases, fiber degradation seems to be enhanced [212], whereas other studies have shown a decrease in the digestion of cell wall carbohydrates in the rumen. As a result, EO effects on the extent of degradation of feed in the rumen may be highly variable, depending on the source and type of EO, dose used, and methodology used in each study. Several studies have evaluated the effects of EO on ruminal fermentation in vitro. An increase in the culture medium’s pH has been observed, although the difference observed depends on the EO added, the dose applied, and the buffer used in the medium. In general, pH is increased to a greater extent, with higher concentrations of phenols or cinnamaldehyde in the medium. This effect can be associated with the decrease in volatile fatty acid concentration in the medium, regardless of the EO tested. Although in some in vitro studies, a slight reduction in VFA production was observed when EO was added to the medium, other reports show little or no EO effect on VFA production or concentration. The VFA profile (molar proportion of each VFA) is a suitable indicator of any shift in the rumen fermentation pattern occurring in response to EO. Again, the literature results are highly diverging, although most commonly, effects showing a decrease in acetate and a trend to increase propionate and butyrate [212,214] have been reported. As a result, it is commonly observed that EO supplementation is related to a decrease in the acetate to propionate ratio in the rumen digesta [214,215]. Figure 5 summarizes the main effects of EOs in ruminal fermentation. 

EOs may also affect the degradation and metabolism of feed proteins in the rumen by reducing the reactions of deamination of amino acids, thus decreasing ammonia production. In this sense, EOs could selectively prevent the proliferation of hyper-ammonia-producing bacteria, responsible for these reactions [217]. Only phenols and cinnamaldehyde decrease the concentration of ammonia-nitrogen (N-NH_3_) [218].

Due to its antimicrobial activity, a commonly recognized potential activity of EOs is the inhibiting effect on methanogenic archaea responsible for methane production. This is one of EOs’ most widely recognized effects on ruminal fermentation and has been extensively explored (mainly in vitro) to reduce methane emissions from ruminants. Methane represents a loss of gross energy from feed and is considered a greenhouse gas. Thus, any reduction in methane derived from ruminal fermentation may enhance feed efficiency and reduce ruminant farming’s environmental impact. Nevertheless, although the reducing effects on ruminal methanogenesis have been shown by several studies testing a diversity of EO [216,219,220], some results are also questionable, as Jafari showed in his review on the effects of oregano, carvacrol, and thymol on ruminal methanogenesis [211]. 

Furthermore, EO can also affect some ruminal microorganisms involved in the bio-hydrogenation pathways, thus causing a modification of fatty acid profile/Polyunsaturated fatty acids (PUFAs) in ruminal digesta when EOs are fed to ruminants. Rosemary and thyme by-products seem to have a potential to increase the unsaturated FA content (particularly n-3 FA) in animal products due to their content in polyphenol compounds, which may inhibit targeted ruminal bacteria involved in biohydrogenation [221]. It must be considered, however, that different compounds and proportions are found in other EOs, blends, so to better describe the effects caused by these products at a ruminal level, it is desirable to carry out studies with pure compounds (e.g., carvacrol, thymol, eugenol) added at different levels, to better understand the modifications produced on ruminal microbiota and bio-hydrogenation pathways. 

Summing up, the antimicrobial activity of some of the compounds contained in EO may affect the rumen microbiota. This may cause a shift in the ruminal fermentation pattern. Thus, in the derived digestion end-products that are either absorbed and used by the animal or disposed of as waste metabolites (methane or urea from ammonia), any effects of EO (at a given dose) must be the most selective as possible. Any broad and non-specific activity will cause a general inhibition of ruminal fermentation with no beneficial impacts on the ruminant. Any success on any of the targeted activities may significantly impact animal metabolism, feed efficiency, product quality, and environmental impact. However, the literature results show that these effects are highly dependent on EO type, source, applied dose, and the mixture of EOs. Effects can also be transient due to an adaptation of the microbiota to survive EOs’ antimicrobial effects. Thus, although results may suggest some tendencies, not entirely conclusive outcomes can be drawn from the available information. As an example, although effects from both carvacrol and thymol have been demonstrated in vitro, recently, the authors of Reference [222] have reported that no such effects on ruminal fermentation, animal productivity, and feed efficiency were observed in dairy cows in vivo.

## 10. EOs in Animal Nutrition

The effects of EOs, when supplied in the feed on the meat and milk quality of the animal, are contradictory, probably due to the different compositions of the EOs administered in each study. However, there is a consensus that low doses of EOs when feeding between 1.33 and 4 g/animal/day, may improve, in some cases, meat quality and shelf life (lessened color degradation, increased antioxidant activity, and decreased lipid oxidation in the meat). A dose of 3.5 g/animal/day could be recommended in feedlot animals [223,224]. These low doses have shown some antioxidant activity, which may reduce the lipid peroxidation of meat (and improve color) and the oxidation proteases such as calpain, thus enhancing the tenderness of meat [225]. 

An extensive review describing the mechanisms of actions of EOs to improve meat quality of lambs has been recently published by Garcia-Galicia et al. [226].

However, greater doses could have a pro-oxidant effect on the animal, with detrimental consequences on health status and product quality. This is because high doses of EO can permeabilize the mitochondria, thus changing electrons’ flow and producing more free radicals, such as reactive oxygen species (ROS) that oxidize lipids and proteins. Some metabolites might be accumulated at the meat level, since supplementing lambs with rosemary EOs improved the sensory properties of lamb’s meat (e.g., flavor and overall acceptability were increased) [221]. However, it is interesting to note that most of the studies published describing the effects of EOs on meat quality when being fed to animals do not evaluate the transmission of metabolites/compounds to the product, which can be different according to the absorption distribution metabolism and excretion (ADME) of the various components included in the composition of the EO administered. Therefore, many of these studies cannot attribute the effects observed in meat quality to direct or indirect effects caused by the different compounds of EOs at the meat level. This approach should be addressed in the future for a better comprehension of the mechanisms of action unchained by EOs at the meat level.

Similar doses of EOs have been suggested when feeding dairy ruminants to improve feed efficiency and milk quality. For example, Al-Suwaiegh et al. have recently described improvements in milk yield, milk total bacterial count, milk somatic cell count, and feed efficiency when feeding the lowest dose (2.5 g/head/day vs 5 g/head/day) of an EOs blend of clove, oregano, and juniper in equal proportions to early lactating Holstein dairy cows. They attributed these effects, at least partially, to lack of inhibition of the rumen microbial populations by the lowest EO dose [227]. 

A recent study done by El-Essawy et al. evaluated the effect of a mixture of EOs (anise, clove, and thyme) on eight lactating shame goats. Supplementation of these EOs improved digestibility with no effect on intake and milk yield. They found that EOs’ supplementation improved the fermentation (digestibility of organic matter, ether extract, and acid detergent fiber was higher). Adding the EO increased milk fat yield, fat content, and the concentrations of unsaturated and monounsaturated fatty acids (FA) compared with the control diet [228]. Benchaar found that dairy cows’ feed added with *Thymus* EO failed to improve the fermentation [222]. Soltan et al. studied the effect of a microencapsulated mixture of EOs containing cinnamaldehyde, eugenol, carvacrol, and capsicum oleoresin on Santa Inês sheep. They found that the feed implementation of this mixture decreases methane production [229].

Adding EOs to animal feed could affect the quality of fat in milk and meat, by inhibiting bacteria responsible for the bio-hydrogenation of unsaturated fat. It was also proven that supplementation in anise, nails clove, and juniper EOs improves the concentration of Conjugated Linoleic Acids and omega 3 in dairy goats [230]. Kholif et al. demonstrated that adding *Capsicum* and *Thymus* EO with fibrolytic enzyme (obtained from *Aspergillus niger*) to Farafra ewes’ feeding increased the milk production, the feed efficiency, and the fat content in the milk. This mixture enhances the fiber digestibility by increasing the number of cellulolytic bacteria, causing an increase in the fat quantity. It also minimizes the energy and protein losses, causing an increase in milk production [231]. 

## 11. EOs in Food Products 

To enhance the shelf life of food products and conserve their sensory characteristics, EOs could be added directly to food or in active food packaging. Researchers have been studying this phenomenon for years, and they have proven its effectiveness in vegetables, fish, dairy products, poultry, and beef packages. The EO is incorporated into a polymer matrix which is used to pack food. The migration of EO into food was studied to ensure the sensory stability of the food product. Table 4 lists five recent studies where the EOs were used in the food packaging.

Several papers have demonstrated the inhibitory effect of EOs to extend the shelf life of beef burgers by controlling the proliferation of pathogenic bacteria through antimicrobial activity. This is due to hydroxyl groups’ presence in phenolic compounds (e.g., thymol, eugenol, carvacrol) which can degrade the bacterial cell wall, disturbing the phospholipid bilayer of the cytoplasmic membrane, causing leakage and cell death. They can further disrupt the proton motive force, electron flow, and active transport, coagulate the cell contents, impair enzymes involved in the energy regulation and synthesis of structural components [147], or inactivate or destroy genetic material, increasing the EOs’ antimicrobial activities. 

Meat and poultry products are also sensitive to oxidation. This process leads to the deterioration of odor, color, and flavors in these products [237]. Adding EOs to the meat products enhances their shelf life by inhibiting their oxidation: Adding rosemary EO on frankfurters significantly decreases the number of carbonyls originated from protein. It also reduces the color change, hardness, chewiness, and frankfurters’ adhesiveness during storage in a fridge [238]. Another study shows that *Thymbra spicata* EO was more effective than synthetic antioxidants on Turkish dry-fermented sausage [239]. Adding oregano EO in salchichon (fermented dry Spanish sausages) led to an increase in the quantity of unsaturated fatty acids without affecting the lipolysis process or the final product’s sensory attributes [240]. 

However, this beneficial activity of EOs when added to meat can be reduced when there is a high content of fat, for example [241]. An extensive review of the antimicrobial effects of EOs when added to meat was published by Pateiro et al. [242], compiling the different extraction protocols for EOs.

EOs could be added to edible oils and fat to block unsaturated oils’ oxidative reaction, causing unpleasant taste and flavor production. The oxidation of fat could also produce some toxic compounds, such as aliphatic and aromatic ketones, carboxylic acids, aldehydes, esters, sulfoxides, phenols, anhydrides, quinones, and aliphatic and aromatic alcohols [243]. Adding oregano EO in a combination of saturated and unsaturated fatty acids inhibited the oxidation process in the presence of light for one year [244]. The same results were observed when blackseed EO (*Nigella sativa* L.) was added to sunflower oil [203], clove and thyme EOs to cottonseed oil [245], and zenyan EO to mayonnaise [246].

EOs could also be added in dairy products: (i) Adding *Thymus capitatus* EO to raw milk could improve its pasteurization and improve its conservation by delaying its spoilage [247]. (ii) The incorporation of 250 μL/kg of EO such as cinnamon, clove, and rosemary in concentrated yoghurt could also increase the product’s shelf life [248]. (iii) EOs could also be added to ice cream: Adding lemon peel and lavender EOs inhibits the development of *Salmonella*, *E. coli*, *Listeria,* and *S. Aureus* [249]. (iv) *Thymus* and cumin EOs could be added to butter to prevent its deterioration during room-temperature storage [250].

## 12. Conclusions

The health status of animals and the quality of the final products are heavily affected by the adverse effects caused by the oxidative stress [16,251,252]. EOs isolated from plants display biological properties with antioxidant, anti-inflammatory, and antimicrobial activities [115,156], leading to environmentally friendly technologies and limiting antimicrobial resistance. Their robust antioxidant properties can be attributed, at least in part, to their richness in terms of phenolic derivatives acting by preventing lipid peroxidation, scavenging free radicals, or, and in very few cases, chelating metal ions [156,157]. 

Due to the rising concerns on the safety of their long-term consumption, EOs as food additives protecting against foodborne pathogens have come to the forefront of this field of research. Moreover, chemical preservatives cannot eliminate several pathogenic bacteria on food products, such as *Listeria monocytogenes,* or delay the growth of spoilage microorganisms. EOs may play that role. They are also environmentally friendly and have the advantage of being better tolerated in the human body, usually with fewer side effects [10]. 

Effectiveness of EOs in extending the shelf life and oxidative stability of dairy and meat products was proven [147]. However, some issues are raised. EOs can negatively alter the product’s organoleptic properties due to their intense aroma, which may affect the consumer acceptability [253]. Moreover, the possible interaction between EOs’ components and constituents, such as fats, carbohydrates, protein, and salts, should be considered. These reactions may reduce the antioxidant activity of these natural products [147]. The synergistic effect of the combination of different EOs could address this issue since it allows lower concentrations of EOs combined with other compounds, thus enhancing their antioxidant effect. 

Using EOs to promote animal health and welfare from a One Health perspective is very important (Figure 6). Studies in controlled conditions mostly agree that EOs affect the immune system by activating the arachidonic metabolism or cytokine production and/or the modulation of pro-inflammatory gene expression by regulating the NF-κB inflammatory pathway and MAPK signaling pathway [136]. Macrophage phagocytosis and oxidative burst have also been considered as additional potential mechanisms modulated by the action of EOs [137]. Nevertheless, the effects were mostly demonstrated under in vitro conditions. More well-designed in vitro and in vivo studies, involving a normalization of dose, taking into account environmental factors such as the type of the diet, breed, and age, will allow a better understanding of EOs’ biological activities under several environmental conditions. In addition, the activities’ molecular bases have not been primarily investigated in depth in this field and may change depending on the specific EO used and the type and severity of the disease. Thus, considering the multitude of components of EOs and the spectrum of possible activities, there is still a vast amount unknown about their actual effects on animal health and particularly in ruminants.

Emerging studies in cattle indicate an improvement in ruminant performance and feed efficiency when EOs are given in the diet [228,230,231], which may add perspective in using EOs in animal nutrition. These positive effects might be partially due to the modulation exerted on the ruminal microbiota, increasing the fiber’s digestibility to produce volatile fatty acids and synthesize microbial protein as an energy and protein supply ruminant, respectively [254,255]. Recent studies with thymol supplementation contrasted with the claimed in vitro antimicrobial activity of EOs: no positive effects on rumen metabolism (i.e., N and VFA) nor beneficial effects on rumen fermentation, nutrient utilization, and milk performance in dairy cows were observed, indicating the complexity of each EO effect per se. Care should be taken before assuming that the property of EOs is simply that of one characteristic component. Identifying a leading beneficial compound and the elucidation of the exact mechanism of action is complicated due to a plethora of compounds with different molecular structures, and antioxidant and antimicrobial activities. More in-depth knowledge of the mechanism of action and effects of individual compounds would allow the formulation of compounds with optimized efficacy and EOs’ inclusion in the diet.

Given the plasticity of the gut and the DOHaD concept’s microbiome, the potential use of EOs in early nutrition programs (gestation and lactation) would be worthwhile. Early EO supplementation may improve the immunity responses with beneficial effects in feed efficiency and later-life performance [256,257,258]. This strategy might be considerably relevant. It can reduce both the administration time (limited only to the first two months of life instead of during the lactation) and the dose needed to positively affect calves at adult age. A reduction in the total cost of the treatment might be expected. However, attention should be paid regarding the doses given to the fetus and newborn due to the unknown potential toxicity effects of EOs when given during early life.

In summary, our review identified EOs as a potential future therapeutic option in both ruminant and monogastric husbandries. EOs are unlikely to replace antimicrobials, but they can at least be used as a complementary treatment. The results of this review provide support for a need for additional in vitro and in vivo research focused on EOs as antioxidant, anti-inflammatory, and antimicrobial agents for recently emerging and challenging diseases in livestock and to improve animal-derived products. While many peer-reviewed studies about EOs were performed in humans and experimental models, more research is needed to evaluate the potential of EOs for the treatment of farm animals. 

Regarding feeding, hygiene, and conservation, EOs are part of a sustainable, natural option for improving animal health and animal product-derived food, reducing the use of antimicrobials in livestock farming. 

We hope that the present review will stimulate further in vitro and in vivo research focused on EOs’ effects among livestock management practice and diseases prevention, opening new avenues in this field and providing guidance on EOs as new challenging molecules in livestock.

## Figures and Tables

**Figure 1 antioxidants-10-00330-f001:**
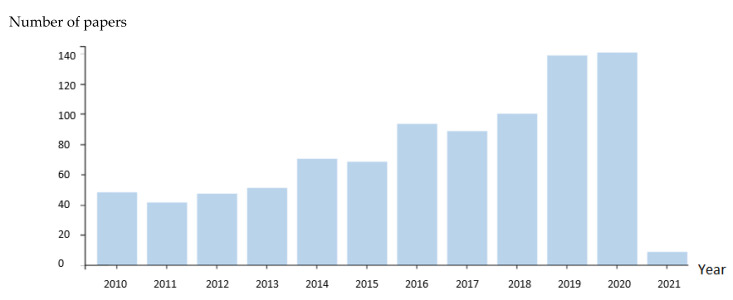
Distribution graph of the number of the articles containing essential oils (EOs) and a livestock animal or an animal food product according to the year of publication (2010–2021).

**Figure 2 antioxidants-10-00330-f002:**
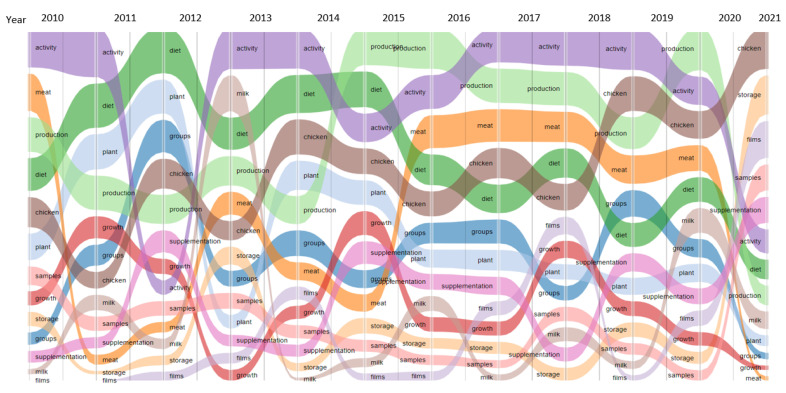
“Epic epoch” graph allowing to visualize the first 5 most used keywords in each article for each year, from 2010 to 2021.

**Figure 3 antioxidants-10-00330-f003:**
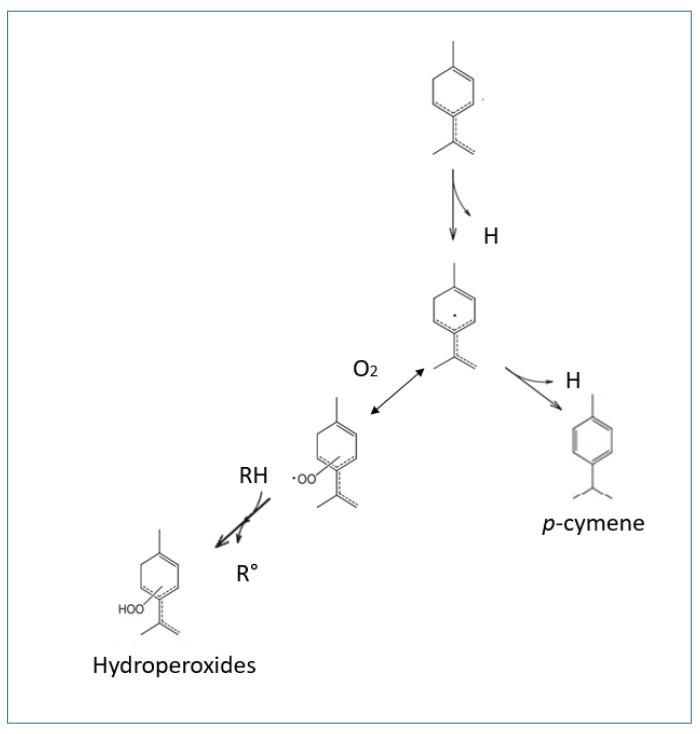
Oxidation of terpenoids adapted from Reference [95].

**Figure 4 antioxidants-10-00330-f004:**
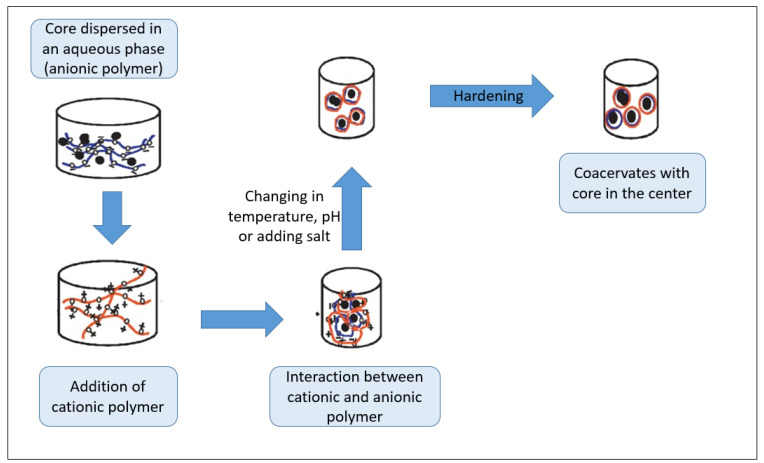
Coacervation process adapted from Reference [105]. The bioactive molecule to be encapsulated (core) is first mixed with one polymer before adding the oppositely charged polymer. The complex coacervation process is generally spontaneous or induced by altering the physicochemical parameters (pH, temperature, ionic strength, etc.). This process leads to the formation of micrometric droplets (5–10 µm) with entrapped bioactive molecules. At the end, stabilization can be performed using chemical or physical treatments.

**Figure 5 antioxidants-10-00330-f005:**
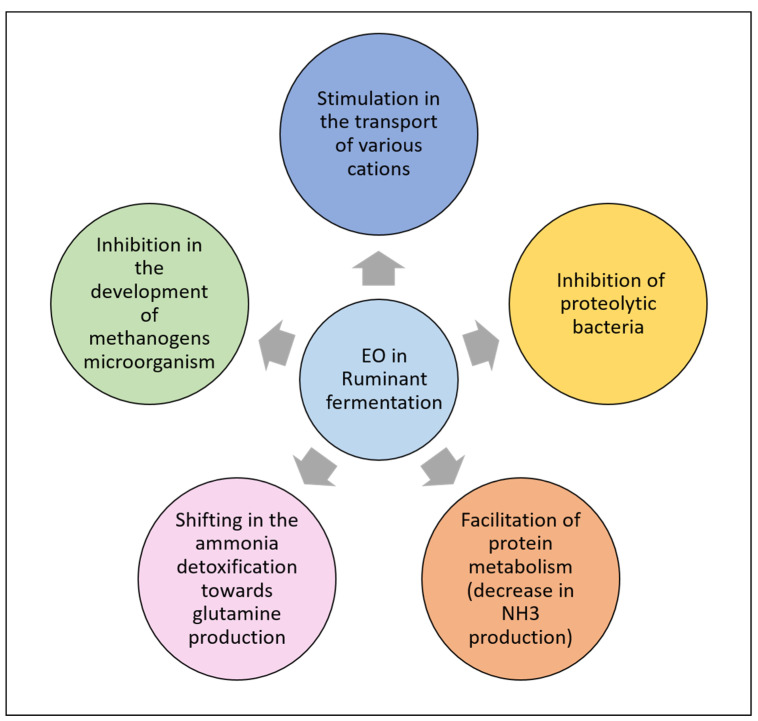
Essential oils in ruminant fermentation [216].

**Figure 6 antioxidants-10-00330-f006:**
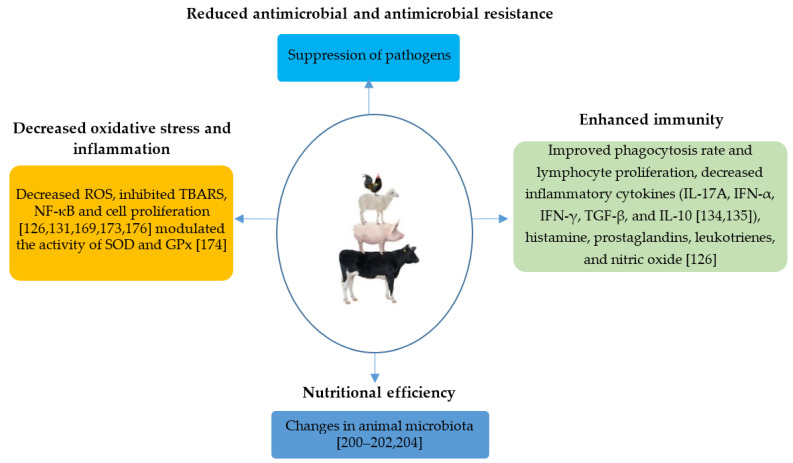
EOs in livestock from a One Health perspective. IL: Interleukin; NF-κB: Nuclear Factor Kappa B; TGF-β: Transforming growth factor-beta; IFN: Interferon; GPx: Glutathione peroxidase; SOD: superoxide dismutase; ROS: Reactive oxygen species, TBARS: Thiobarbituric acid-reactive species.

**Table 1 antioxidants-10-00330-t001:** Major components of selected essential oils (EOs) that exhibit antibacterial properties.

Latin Name of Plant Source	Major Components	Approximate % Composition	References
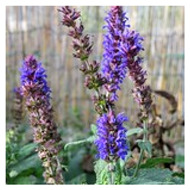 *Salvia officinalis* L. Credit: Dusan Baksa	Camphor α-Pinene β-Pinene 1,8-Cineole α-Thujone	6–15% 4–5% 2–10% 6–14% 20–42%	[68]
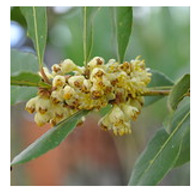 *Laurus nobilis* Credit: Stefano	1,8-Cineol Borneol	51.63–63.19% 5.8–12.80%	[69]
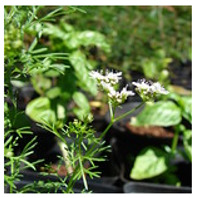 *Coriandrum sativum* Credit: Forest & Kim Starr	Linalool	40–70%	[70,71]
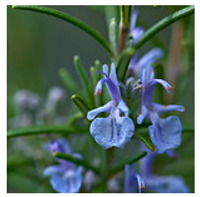 *Rosmarinus officinalis* Credit: Peter Stenzel	α-Pinene Bornyl acetate Camphor 1,8-Cineole	2–25% 0–17% 2–14% 3–89%	[72,73,74]
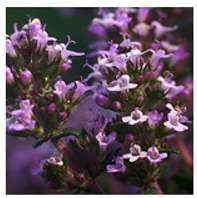 *Thymus vulgaris* Credit: Ralf Wimmer	Thymol Carvacrol δ-Terpinene *p*-Cymene	10–64% 76.1% 2–31% 10–56%	[59,63,66,72,75]
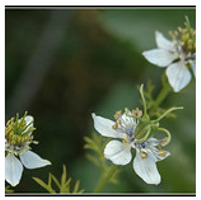 *Nigella sativa* Credit: Eran Finkle	Thymoquinone Carvacrol *p*-Cymene	27–45% 7–11.6% 7–15%	[76]
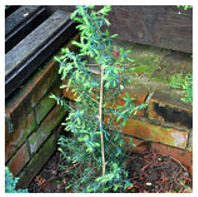 *Juniperus officinalis* Credit: Zoran Radosavljevic	Β-Phellandrene α-Terpinyl acetate α-Pinene α-Phellandreneinene Manonyl oxide	4.9–23.8% 4.6–15.5% 28–60% 0.8–17.7% 0.4–14.4%	[77]
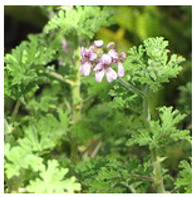 *Pelargonium graveolens* Credit: Forest & Kim Starr	Citronellol Geraniol Linalool Citronellyl formate *p*-Menthone	33.6% 26.8% 10.5% 9.7% 6%	[78]
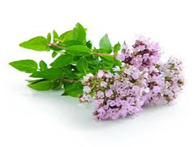 *Origanum vulgare* Credit: Koromelana.yandex.ru	Carvacrol Thymol γ-Terpinene	Trace–80% Trace–64% 2–52%	[40,68,72,79,80,81,82,83]
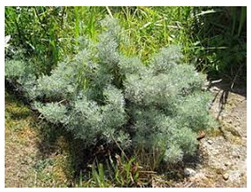 *Artemisia herba alba* Credit: Peganum	Limonene Fenchol α-Thujone Camphor Nordavanone	4–9% 7–14% 11–14% Trace–30% 1–9%	[84]

**Table 2 antioxidants-10-00330-t002:** List of recent studies that tested the antibacterial properties of some EOs against pathogens by identifying the minimum inhibitory concentration (MIC) of EOs.

EOs	Bacteria	MIC	References
*Cymbopogon* spp*. and Cinnamomum verum*	*Escherichia coli*	0.075% to 0.3% (*v*/*v*) and 0.0075% (*v*/*v*), respectively	[116]
*A. sativum*, *C. verum*, *O. basilicum*, *S. aromaticum* and *T. vulgaris*	*Aeromonas hydrophila*, *A. jandaei* and *Citrobacter freundii* isolated from diseased freshwater fish	MIC ≤ 500 µg/mL	[117]
*Lavandula x intermedia*	*Streptococcus agalactiae* and *Candida albicans*	MIC 9–18 µg/mL for *Streptococcus* MIC 9–18 µg/mL for *Candida*	[118]
*Mentha arvensis*	*Streptococcus agalactiae* and *Candida albicans*	MIC 18–36 µg/mL for *Streptococcus* MIC 18–144 µg/mL for *Candida*	[118]
*P. armeniaca*	*M. luteus, S. aureus and E. coli*	MIC 23.4 µg/mL for *M. luteus* MIC 23.4 µg/mL for *S. aureus* MIC 11.7 µg/mL for *E. coli*	[119]
*L. nobilis*	*M. luteus, S. aureus and B. subtilis*	MIC 22.2 µg/mL for *M. luteus* MIC 5.55 µg/mL for *S. aureus* MIC 1.39 µg/mL for *B. subtilis*	[119]
*M. alternifolia*	*K. pneumoniae, P. aerginosa*, and *E coli*	MIC 0.5 and 4 μg/mL	[120]
*T. vulgaris*	*K. pneumoniae**, P. aerginosa*, and *E coli*	MIC 1 to 16 μg/mL	[120]
Garlic, cinnamon, and onion	*L. monocytogenes*	MIC respectively 100, 100, and 25 μg/mL	[121]

**Table 4 antioxidants-10-00330-t004:** Five selected bibliographical studies showing the beneficial use of EOs in different food packaging.

EO	Polymer	Food	Results	Reference
*Laurus nobilis* and *Rosmarinus officinalis*	Polyvinyl alcohol	Chicken breast fillets	Inhibition of the lipid oxidation up to 68%. Beneficial effect on both the pH and color parameters of the fillets during storage.	[232]
Rosemary and cinnamon EO	Polyethylene films	Pacific white shrimp	Prolonged and retained shrimp freshness and shelf life	[233]
Carvacrol, oregano, and cinnamon	Cardboard tray	Cherry tomato	Tomato color and firmness were highly maintained during storage. The shelf life was extended up to 4 days	[234]
Cumin EO	Shahri Balangu seed mucilage	Beef	Expand the shelf life of the beef	[235]
Oregano	Edible coating: mandarin fiber and sodium alginate	Low-fat cheese	Decontamination of external pathogens such as *Staphylococcus aureus*	[236]

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
