# Peer review of "Essential Oils in Livestock: From Health to Food Quality"

_antioxidants, 2021, doi:10.3390/antiox10020330_

Round 1
Reviewer 1 Report
Antioxidants-1113446
Essential oils in livestock: From health to food quality
This paper is a very interesting and well written reviw on essential oils with particular reference to their use as food enhancer in livestock.
The introduction is complete with a brief overview on EO uses, reference to the problem of antibiotic used in animal fooder. The authors deepen that use in animals with interesting reference to potential uses to improve milk and meet quality.
I like the way the authors stress that EO are better better than dried plant material. I appreciate the precise definitions used in the text as there is often confusion on the use of plant derivated volatiles.
The most significant use of EOs in the European Union (EU) is highlighted.
The authors correctly stress that diferent methods give different yeld and composition. Perhaps the dependence on cultivation methods can be discussed (water availability, higness, temperatue and other).
The authors continus with discussion on stability and encapsulation, main properties and present a case study on bovine mastitis. Good one.
table 2 is reductive as it refers to a previous review dating back to 2008. Authors should make a search for new data (there are really a lot of more recent ones)
The paragraph on antiinflammatory properties is not up to date. This paper is a review and must report more references, not simply rely on previous ones (dating back to 2005 and 2010).
Chapters 8, 9 and 10 are OK, very interesting with references up to date
There are several calls for not found references. I checked most of them and found at several sites. Below I list some examples.
line 205 there is a call for ref (69) not found. I traced it at https://onlinelibrary.wiley.com/doi/abs/10.1002/(SICI)1099-1026(199803/04)13:2%3C98::AID-FFJ705%3E3.0.CO;2-B
line 247 there is a call for ref (97) not found. I traced it at
https://onlinelibrary.wiley.com/doi/full/10.1111/1541-4337.12006
in Line 354 there is a call for ref (117) not found. I traced it at (117)
https://www.sciencedirect.com/science/article/abs/pii/S0141813018348967
Line 565 and 565: Ref 158 qnd 159 are OK
again, refs refs 210 and 211 are available on the Web
Reviewer 2 Report
The novelty character of this review should be marked in the aim and in the Conclusion.
The paragraph 2.3. Current use of EOs should be implemented.
A Methodological section should be inserted by describing the bibliographic research criteria and literature workflow including graphical scheme.
Major details in Figure 2 should be given.
The paragraph 6.1. Antimicrobial activity should be implemented including Table
The paragraph 6.2. Anti Inflammatory properties should be implemented and a Table added.
Reviewer 3 Report
The authors present a review on essential oils their possible in animal production and with aspects of the quality of the food produced thereof. They deal with the topics extraction and, composition, of essential oils, encapsulation and major biological activities. Applications in animal husbandry might be their anti-inflammatory activity to treat mastitis, uses as feed additives, modulation of rumen metabolism, and in animal derived foods.
The review is of interest because of its focus on animal production.
Please consider
Lines 65-68 and 68-71: are identical, delete
Line 86: DOHaD or DOHAD?
Line 132: …between them turpentine…
Table 1, Pelargonium graveolens: why are citronellol, geraniol and linalool underlined? Origanum vulgare: γ-terpinene instead of δ-terpinene, Artemisia herba-alba: α-thujone
Line 253: …known as core material, with a protective layer called wall material.
Line 284: …can involve various steps….
Lines 348-350: scientific names in italics
Line 352: deleted the bold text
Line 382: Lavandula angustifolia in italics
Line 392: Format of these reference. This reference is not in the reference list
Line 426: …expression of some pro-inflammatory…
Lines 431 and 436: Minthostachys verticillata in italics
Lines 431-433: How are in this study the results from the mouse model related to ruminants?
Line 437: EO instead of EPO
Line 438, 455-459, 484-485, 498, 504, 512, 532, 533, 537, 538, 573-576, 579, 583, 586, 593, 597-604 and at several instances thereafter in the main text: please set scientific names in italics
Line 499: Sanguisorba minor is a plant containing polyphenols and tannins rather than essential oils
Line 503: The EOs… inhibit respiration….
Lines 568-569: delete: , the most widespread species of Lamiaceae family,
More variable in comparison to what?
- 21, bottom (Line 663): ….compounds may not be inhibitory…
- 24, line 770: …Al-Suwaiegh et al. have recently….
- 25, line above the table: …product. Table lists 5 recent studies…
References:
At many instances: please set scientific names in italics
Ref. 47: Journal title is mentioned twice
Ref. 48 and 51 appear to be identical
Ref. 110 is incomplete
Ref 210: ruminal fermentation
Ref. 219 and 220 are identical
Additionally, a list of abbreviations used might be helpful
